# Bayesian Inference and Learning in Gaussian Process State-Space Models with Particle MCMC

**Roger Frigola[1], Fredrik Lindsten[2], Thomas B. Schön[2,3] and Carl E. Rasmussen[1]**

1. Dept. of Engineering, University of Cambridge, UK, {rf342,cer54}@cam.ac.uk
2. Div. of Automatic Control, Linköping University, Sweden, lindsten@isy.liu.se
3. Dept. of Information Technology, Uppsala University, Sweden, thomas.schon@it.uu.se

## Abstract

State-space models are successfully used in many areas of science, engineering and economics to model time series and dynamical systems. We present a fully Bayesian approach to inference *and learning* (i.e. state estimation and system identification) in nonlinear nonparametric state-space models. We place a Gaussian process prior over the state transition dynamics, resulting in a flexible model able to capture complex dynamical phenomena. To enable efficient inference, we marginalize over the transition dynamics function and, instead, infer directly the joint smoothing distribution using specially tailored Particle Markov Chain Monte Carlo samplers. Once a sample from the smoothing distribution is computed, the state transition predictive distribution can be formulated analytically. Our approach preserves the full nonparametric expressivity of the model and can make use of sparse Gaussian processes to greatly reduce computational complexity.

## 1 Introduction

State-space models (SSMs) constitute a popular and general class of models in the context of time series and dynamical systems. Their main feature is the presence of a latent variable, the *state* $\mathbf{x}_t \in \mathsf{X} \triangleq \mathbb{R}^{n_x}$, which condenses all aspects of the system that can have an impact on its future. A discrete-time SSM with nonlinear dynamics can be represented as

$$\mathbf{x}_{t+1} = f(\mathbf{x}_t, \mathbf{u}_t) + \mathbf{v}_t, \tag{1a}$$

$$\mathbf{y}_t = g(\mathbf{x}_t, \mathbf{u}_t) + \mathbf{e}_t, \tag{1b}$$

where $\mathbf{u}_t$ denotes a known external input, $\mathbf{y}_t$ denotes the measurements, $\mathbf{v}_t$ and $\mathbf{e}_t$ denote i.i.d. noises acting on the dynamics and the measurements, respectively. The function $f$ encodes the dynamics and $g$ describes the relationship between the observation and the unobserved states.

We are primarily concerned with the problem of learning general nonlinear SSMs. The aim is to find a model that can adaptively increase its complexity when more data is available. To this effect, we employ a Bayesian nonparametric model for the dynamics (1a). This provides a flexible model that is not constrained by any limiting assumptions caused by postulating a particular functional form. More specifically, we place a Gaussian process (GP) prior [1] over the unknown function $f$. The resulting model is a generalization of the standard parametric SSM. The functional form of the observation model $g$ is assumed to be known, possibly parameterized by a finite dimensional parameter. This is often a natural assumption, for instance in engineering applications where $g$ corresponds to a sensor model – we typically know what the sensors are measuring, at least up to some unknown parameters. Furthermore, using too flexible models for both $f$ and $g$ can result in problems of non-identifiability.

We adopt a fully Bayesian approach whereby we find a posterior distribution over all the latent entities of interest, namely the state transition function $f$, the hidden state trajectory $\mathbf{x}_{0:T} \triangleq \{\mathbf{x}_i\}_{i=0}^T$

and any hyper-parameter $\boldsymbol{\theta}$ of the model. This is in contrast with existing approaches for using GPs to model SSMs, which tend to model the GP using a finite set of target points, in effect making the model parametric [2]. Inferring the distribution over the state trajectory $p(\mathbf{x}_{0:T} \mid \mathbf{y}_{0:T}, \mathbf{u}_{0:T})$ is an important problem in itself known as *smoothing*. We use a tailored particle Markov Chain Monte Carlo (PMCMC) algorithm [3] to efficiently sample from the smoothing distribution whilst marginalizing over the state transition function. This contrasts with conventional approaches to smoothing which require a fixed model of the transition dynamics. Once we have obtained an approximation of the smoothing distribution, with the dynamics of the model marginalized out, learning the function $f$ is straightforward since its posterior is available in closed form given the state trajectory. Our only approximation is that of the sampling algorithm. We report very good mixing enabled by the use of recently developed PMCMC samplers [4] and the exact marginalization of the transition dynamics.

There is by now a rich literature on GP-based SSMs. For instance, Deisenroth et al. [5, 6] presented refined approximation methods for filtering and smoothing for already learned GP dynamics and measurement functions. In fact, the method proposed in the present paper provides a vital component needed for these inference methods, namely that of learning the GP model in the first place. Turner et al. [2] applied the EM algorithm to obtain a maximum likelihood estimate of parametric models which had the form of GPs where both inputs and outputs were parameters to be optimized. This type of approach can be traced back to [7] where Ghahramani and Roweis applied EM to learn models based on radial basis functions. Wang et al. [8] learn a SSM with GPs by finding a MAP estimate of the latent variables and hyper-parameters. They apply the learning in cases where the dimension of the observation vector is much higher than that of the latent state in what becomes a form of dynamic dimensionality reduction. This procedure would have the risk of overfitting in the common situation where the state-space is high-dimensional and there is significant uncertainty in the smoothing distribution.

## 2 Gaussian Process State-Space Model

We describe the generative probabilistic model of the Gaussian process SSM (GP-SSM) represented in Figure 1b by

$$f(\mathbf{x}_t) \sim \mathcal{GP}\big(m_{\boldsymbol{\theta}_\mathbf{x}}(\mathbf{x}_t), k_{\boldsymbol{\theta}_\mathbf{x}}(\mathbf{x}_t, \mathbf{x}_t')\big), \tag{2a}$$

$$\mathbf{x}_{t+1} \mid \mathbf{f}_t \sim \mathcal{N}(\mathbf{x}_{t+1} \mid \mathbf{f}_t, \mathbf{Q}), \tag{2b}$$

$$\mathbf{y}_t \mid \mathbf{x}_t \sim p(\mathbf{y}_t \mid \mathbf{x}_t, \boldsymbol{\theta}_\mathbf{y}), \tag{2c}$$

and $\mathbf{x}_0 \sim p(\mathbf{x}_0)$, where we avoid notational clutter by omitting the conditioning on the known inputs $\mathbf{u}_t$. In addition, we put a prior $p(\boldsymbol{\theta})$ over the various hyper-parameters $\boldsymbol{\theta} = \{\boldsymbol{\theta}_\mathbf{x}, \boldsymbol{\theta}_\mathbf{y}, \mathbf{Q}\}$. Also, note that the measurement model (2c) and the prior on $\mathbf{x}_0$ can take any form since we do not rely on their properties for efficient inference.

The GP is fully described by its mean function and its covariance function. An interesting property of the GP-SSM is that any *a priori* insight into the dynamics of the system can be readily encoded in the mean function. This is useful, since it is often possible to capture the main properties of the dynamics, e.g. by using a simple parametric model or a model based on first principles. Such

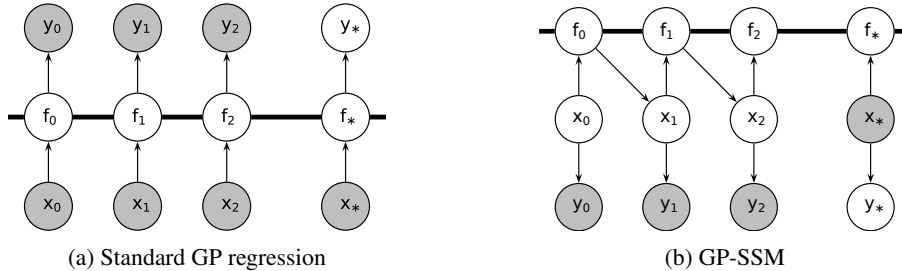

(a) Standard GP regression             (b) GP-SSM

Figure 1: Graphical models for standard GP regression and the GP-SSM model. The thick horizontal bars represent sets of fully connected nodes.

simple models may be insufficient on their own, but useful together with the GP-SSM, as the GP is flexible enough to model complex departures from the mean function. If no specific prior model is available, the linear mean function $m(\mathbf{x}_t) = \mathbf{x}_t$ is a good generic choice. Interestingly, the prior information encoded in this model will normally be more vague than the prior information encoded in parametric models. The measurement model (2c) implicitly contains the observation function $g$ and the distribution of the i.i.d. measurement noise $\mathbf{e}_t$.

## 3 Inference over States and Hyper-parameters

Direct learning of the function $f$ in (2a) from input/output data $\{\mathbf{u}_{0:T-1}, \mathbf{y}_{0:T}\}$ is challenging since the states $\mathbf{x}_{0:T}$ are not observed. Most (if not all) previous approaches attack this problem by reverting to a parametric representation of $f$ which is learned alongside the states. We address this problem in a fundamentally different way by marginalizing out $f$, allowing us to respect the nonparametric nature of the model. A challenge with this approach is that marginalization of $f$ will introduce dependencies across time for the state variables that lead to the loss of the Markovian structure of the state-process. However, recently developed inference methods, combining sequential Monte Carlo (SMC) and Markov chain Monte Carlo (MCMC) allow us to tackle this problem. We discuss marginalization of $f$ in Section 3.1 and present the inference algorithms in Sections 3.2 and 3.3.

### 3.1 Marginalizing out the State Transition Function

Targeting the joint posterior distribution of the hyper-parameters, the latent states *and* the latent function $f$ is problematic due to the strong dependencies between $\mathbf{x}_{0:T}$ and $f$. We therefore marginalize the dynamical function from the model, and instead target the distribution $p(\boldsymbol{\theta}, \mathbf{x}_{0:T} \mid \mathbf{y}_{1:T})$ (recall that conditioning on $\mathbf{u}_{0:T-1}$ is implicit). In the MCMC literature, this is referred to as collapsing [9]. Hence, we first need to find an expression for the marginal prior $p(\boldsymbol{\theta}, \mathbf{x}_{0:T}) = p(\mathbf{x}_{0:T} \mid \boldsymbol{\theta})p(\boldsymbol{\theta})$. Focusing on $p(\mathbf{x}_{0:T} \mid \boldsymbol{\theta})$ we note that, although this distribution is not Gaussian, it can be represented as a product of Gaussians. Omitting the dependence on $\boldsymbol{\theta}$ in the notation, we obtain

$$p(\mathbf{x}_{1:T} \mid \boldsymbol{\theta}, \mathbf{x}_0) = \prod_{t=1}^{T} p(\mathbf{x}_t \mid \boldsymbol{\theta}, \mathbf{x}_{0:t-1}) = \prod_{t=1}^{T} \mathcal{N}\big(\mathbf{x}_t \mid \boldsymbol{\mu}_t(\mathbf{x}_{0:t-1}), \boldsymbol{\Sigma}_t(\mathbf{x}_{0:t-1})\big), \qquad (3a)$$

with

$$\boldsymbol{\mu}_t(\mathbf{x}_{0:t-1}) = \mathbf{m}_{t-1} + \mathbf{K}_{t-1,0:t-2}\widetilde{\mathbf{K}}_{0:t-2}^{-1}(\mathbf{x}_{1:t-1} - \mathbf{m}_{0:t-2}), \qquad (3b)$$

$$\boldsymbol{\Sigma}_t(\mathbf{x}_{0:t-1}) = \widetilde{\mathbf{K}}_{t-1} - \mathbf{K}_{t-1,0:t-2}\widetilde{\mathbf{K}}_{0:t-2}^{-1}\mathbf{K}_{t-1,0:t-2}^{\top} \qquad (3c)$$

for $t \geq 2$ and $\boldsymbol{\mu}_1(\mathbf{x}_0) = \mathbf{m}_0$, $\boldsymbol{\Sigma}_1(\mathbf{x}_0) = \widetilde{\mathbf{K}}_0$. Equation (3) follows from the fact that, once conditioned on $\mathbf{x}_{0:t-1}$, a one-step prediction for the state variable is a standard GP prediction. Here, we have defined the mean vector $\mathbf{m}_{0:t-1} \triangleq \begin{bmatrix} m(\mathbf{x}_0)^{\top} & \dots & m(\mathbf{x}_{t-1})^{\top} \end{bmatrix}^{\top}$ and the $(n_x t) \times (n_x t)$ positive definite matrix $\mathbf{K}_{0:t-1}$ with block entries $[\mathbf{K}_{0:t-1}]_{i,j} = k(\mathbf{x}_{i-1}, \mathbf{x}_{j-1})$. We use two sets of indices, as in $\mathbf{K}_{t-1,0:t-2}$, to refer to the off-diagonal blocks of $\mathbf{K}_{0:t-1}$. We also define $\widetilde{\mathbf{K}}_{0:t-1} = \mathbf{K}_{0:t-1} + \mathbf{I}_t \otimes \mathbf{Q}$. We can also express (3a) more succinctly as,

$$p(\mathbf{x}_{1:t} \mid \boldsymbol{\theta}, \mathbf{x}_0) = |(2\pi)^{n_x t}\widetilde{\mathbf{K}}_{0:t-1}|^{-\frac{1}{2}} \exp(-\frac{1}{2}(\mathbf{x}_{1:t} - \mathbf{m}_{0:t-1})^{\top}\widetilde{\mathbf{K}}_{0:t-1}^{-1}(\mathbf{x}_{1:t} - \mathbf{m}_{0:t-1})). \qquad (4)$$

This expression looks very much like a multivariate Gaussian density function. However, we emphasize that this is not the case since both $\mathbf{m}_{0:t-1}$ and $\widetilde{\mathbf{K}}_{0:t-1}$ depend (nonlinearly) on the argument $\mathbf{x}_{1:t}$. In fact, (4) will typically be very far from Gaussian.

### 3.2 Sequential Monte Carlo

With the prior (4) in place, we now turn to posterior inference and we start by considering the joint smoothing distribution $p(\mathbf{x}_{0:T} \mid \boldsymbol{\theta}, \mathbf{y}_{0:T})$. The sequential nature of the proposed model suggests the use of SMC. Though most well known for filtering in Markovian SSMs – see [10, 11] for an introduction – SMC is applicable also for non-Markovian latent variable models. We seek to approximate the sequence of distributions $p(\mathbf{x}_{0:t} \mid \boldsymbol{\theta}, \mathbf{y}_{0:t})$, for $t = 0, \dots, T$. Let $\{\mathbf{x}_{0:t-1}^i, \mathbf{w}_{t-1}^i\}_{i=1}^{N}$

be a collection of weighted particles approximating $p(\mathbf{x}_{0:t-1} \mid \boldsymbol{\theta}, \mathbf{y}_{0:t-1})$ by the empirical distribution, $\widehat{p}(\mathbf{x}_{0:t-1} \mid \boldsymbol{\theta}, \mathbf{y}_{0:t-1}) \triangleq \sum_{i=1}^{N} \mathbf{w}_{t-1}^i \delta_{\mathbf{x}_{0:t-1}^i}(\mathbf{x}_{0:t-1})$. Here, $\delta_{\mathbf{z}}(\mathbf{x})$ is a point-mass located at $\mathbf{z}$. To propagate this sample to time $t$, we introduce the auxiliary variables $\{\mathbf{a}_t^i\}_{i=1}^{N}$, referred to as *ancestor indices*. The variable $\mathbf{a}_t^i$ is the index of the ancestor particle at time $t-1$, of particle $\mathbf{x}_t^i$. Hence, $\mathbf{x}_t^i$ is generated by first sampling $\mathbf{a}_t^i$ with $\mathbb{P}(\mathbf{a}_t^i = j) = \mathbf{w}_{t-1}^j$. Then, $\mathbf{x}_t^i$ is generated as,

$$\mathbf{x}_t^i \sim p(\mathbf{x}_t \mid \boldsymbol{\theta}, \mathbf{x}_{0:t-1}^{\mathbf{a}_t^i}, \mathbf{y}_{0:t}), \tag{5}$$

for $i = 1, \ldots, N$. The particle trajectories are then augmented according to $\mathbf{x}_{0:t}^i = \{\mathbf{x}_{0:t-1}^{\mathbf{a}_t^i}, \mathbf{x}_t^i\}$. Sampling from the one-step predictive density is a simple (and sensible) choice, but we may also consider other proposal distributions. In the above formulation the resampling step is implicit and corresponds to sampling the ancestor indices (cf. the auxiliary particle filter, [12]). Finally, the particles are weighted according to the measurement model, $\mathbf{w}_t^i \propto p(\mathbf{y}_t \mid \boldsymbol{\theta}, \mathbf{x}_t^i)$ for $i = 1, \ldots, N$, where the weights are normalized to sum to 1.

### 3.3  Particle Markov Chain Monte Carlo

There are two shortcomings of SMC: *(i)* it does not handle inference over hyper-parameters; *(ii)* despite the fact that the sampler targets the joint smoothing distribution, it does in general not provide an accurate approximation of the full joint distribution due to *path degeneracy*. That is, the successive resampling steps cause the particle diversity to be very low for time points $t$ far from the final time instant $T$.

To address these issues, we propose to use a particle Markov chain Monte Carlo (PMCMC, [3, 13]) sampler. PMCMC relies on SMC to generate samples of the highly correlated state trajectory within an MCMC sampler. We employ a specific PMCMC sampler referred to as particle Gibbs with ancestor sampling (PGAS, [4]), given in Algorithm 1. PGAS uses Gibbs-like steps for the state trajectory $\mathbf{x}_{0:T}$ and the hyper-parameters $\boldsymbol{\theta}$, respectively. That is, we sample first $\mathbf{x}_{0:T}$ given $\boldsymbol{\theta}$, then $\boldsymbol{\theta}$ given $\mathbf{x}_{0:T}$, etc. However, the full conditionals are not explicitly available. Instead, we draw samples from specially tailored Markov kernels, leaving these conditionals invariant. We address these steps in the subsequent sections.

---

**Algorithm 1** Particle Gibbs with ancestor sampling (PGAS)

1. Set $\boldsymbol{\theta}[0]$ and $\mathbf{x}_{1:T}[0]$ arbitrarily.
2. **For $\ell \geq 1$ do**
   (a) Draw $\boldsymbol{\theta}[\ell]$ conditionally on $\mathbf{x}_{0:T}[\ell - 1]$ and $\mathbf{y}_{0:T}$ as discussed in Section 3.3.2.
   (b) Run CPF-AS (see [4]) targeting $p(\mathbf{x}_{0:T} \mid \boldsymbol{\theta}[\ell], \mathbf{y}_{0:T})$, conditionally on $\mathbf{x}_{0:T}[\ell - 1]$.
   (c) Sample $k$ with $\mathbb{P}(k = i) = w_T^i$ and set $\mathbf{x}_{1:T}[\ell] = \mathbf{x}_{1:T}^k$.
3. **end**

---

#### 3.3.1  Sampling the State Trajectories

To sample the state trajectory, PGAS makes use of an SMC-like procedure referred to as a conditional particle filter with ancestor sampling (CPF-AS). This approach is particularly suitable for non-Markovian latent variable models, as it relies only on a forward recursion (see [4]). The difference between a standard particle filter (PF) and the CPF-AS is that, for the latter, one particle at each time step is specified *a priori*. Let these particles be denoted $\widetilde{\mathbf{x}}_{0:T} = \{\widetilde{\mathbf{x}}_0, \ldots, \widetilde{\mathbf{x}}_T\}$. We then sample according to (5) only for $i = 1, \ldots, N-1$. The $N$th particle is set deterministically: $\mathbf{x}_t^N = \widetilde{\mathbf{x}}_t$. To be able to construct the $N$th particle trajectory, $\mathbf{x}_t^N$ has to be associated with an ancestor particle at time $t-1$. This is done by sampling a value for the corresponding ancestor index $\mathbf{a}_t^N$. Following [4], the ancestor sampling probabilities are computed as

$$\widetilde{\mathbf{w}}_{t-1|T}^i \propto \mathbf{w}_{t-1}^i \frac{p(\{\mathbf{x}_{0:t-1}^i, \widetilde{\mathbf{x}}_{t:T}\}, \mathbf{y}_{0:T})}{p(\mathbf{x}_{0:t-1}^i, \mathbf{y}_{0:t-1})} \propto \mathbf{w}_{t-1}^i \frac{p(\{\mathbf{x}_{0:t-1}^i, \widetilde{\mathbf{x}}_{t:T}\})}{p(\mathbf{x}_{0:t-1}^i)} = \mathbf{w}_{t-1}^i p(\widetilde{\mathbf{x}}_{t:T} \mid \mathbf{x}_{0:t-1}^i). \tag{6}$$

where the ratio is between the unnormalized target densities up to time $T$ and up to time $t-1$, respectively. The second proportionality follows from the mutual conditional independence of the observations, given the states. Here, $\{\mathbf{x}_{0:t-1}^i, \widetilde{\mathbf{x}}_{t:T}\}$ refers to a path in $\mathsf{X}^{T+1}$ formed by concatenating

the two partial trajectories. The above expression can be computed by using the prior over state trajectories given by (4). The ancestor sampling weights $\{\widetilde{\mathbf{w}}_{t-1|T}^i\}_{i=1}^N$ are then normalized to sum to 1 and the ancestor index $\mathbf{a}_t^N$ is sampled with $\mathbb{P}(\mathbf{a}_t^N = j) = \mathbf{w}_{t-1|t}^j$.

The conditioning on a prespecified collection of particles implies an invariance property in CPF-AS, which is key to our development. More precisely, given $\widetilde{\mathbf{x}}_{0:T}$ let $\widetilde{\mathbf{x}}_{0:T}'$ be generated as follows:

1. Run CPF-AS from time $t = 0$ to time $t = T$, conditionally on $\widetilde{\mathbf{x}}_{0:T}$.
2. Set $\widetilde{\mathbf{x}}_{0:T}'$ to one of the resulting particle trajectories according to $\mathbb{P}(\widetilde{\mathbf{x}}_{0:T}' = \mathbf{x}_{0:T}^i) = \mathbf{w}_T^i$.

For any $N \geq 2$, this procedure defines an ergodic Markov kernel $M_{\boldsymbol{\theta}}^N(\widetilde{\mathbf{x}}_{0:T}' \mid \widetilde{\mathbf{x}}_{0:T})$ on $\mathsf{X}^{T+1}$, leaving the *exact* smoothing distribution $p(\mathbf{x}_{0:T} \mid \boldsymbol{\theta}, \mathbf{y}_{0:T})$ invariant [4]. Note that this invariance holds for any $N \geq 2$, i.e. the number of particles that are used only affect the mixing rate of the kernel $M_{\boldsymbol{\theta}}^N$. However, it has been experienced in practice that the autocorrelation drops sharply as $N$ increases [4, 14], and for many models a moderate $N$ is enough to obtain a rapidly mixing kernel.

### 3.3.2 Sampling the Hyper-parameters

Next, we consider sampling the hyper-parameters given a state trajectory and sequence of observations, i.e. from $p(\boldsymbol{\theta} \mid \mathbf{x}_{0:T}, \mathbf{y}_{0:T})$. In the following, we consider the common situation where there are distinct hyper-parameters for the likelihood $p(\mathbf{y}_{0:T} \mid \mathbf{x}_{0:T}, \boldsymbol{\theta}_\mathbf{y})$ and for the prior over trajectories $p(\mathbf{x}_{0:T} \mid \boldsymbol{\theta}_\mathbf{x})$. If the prior over the hyper-parameters factorizes between those two groups we obtain $p(\boldsymbol{\theta} \mid \mathbf{x}_{0:T}, \mathbf{y}_{0:T}) \propto p(\boldsymbol{\theta}_\mathbf{y} \mid \mathbf{x}_{0:T}, \mathbf{y}_{0:T}) \, p(\boldsymbol{\theta}_\mathbf{x} \mid \mathbf{x}_{0:T})$. We can thus proceed to sample the two groups of hyper-parameters independently. Sampling $\boldsymbol{\theta}_\mathbf{y}$ will be straightforward in most cases, in particular if conjugate priors for the likelihood are used. Sampling $\boldsymbol{\theta}_\mathbf{x}$ will, nevertheless, be harder since the covariance function hyper-parameters enter the expression in a non-trivial way. However, we note that once the state trajectory is fixed, we are left with a problem analogous to Gaussian process regression where $\mathbf{x}_{0:T-1}$ are the inputs, $\mathbf{x}_{1:T}$ are the outputs and $\mathbf{Q}$ is the likelihood covariance matrix. Given that the latent dynamics can be marginalized out analytically, sampling the hyper-parameters with slice sampling is straightforward [15].

## 4 A Sparse GP-SSM Construction and Implementation Details

A naive implementation of the CPF-AS algorithm will give rise to $\mathcal{O}(T^4)$ computational complexity, since at each time step $t = 1, \ldots, T$, a matrix of size $T \times T$ needs to be factorized. However, it is possible to update and reuse the factors from the previous time step, bringing the total computational complexity down to the familiar $\mathcal{O}(T^3)$. Furthermore, by introducing a sparse GP model, we can reduce the complexity to $\mathcal{O}(M^2 T)$ where $M \ll T$. In Section 4.1 we introduce the sparse GP model and in Section 4.2 we provide insight into the efficient implementation of both the vanilla GP and the sparse GP.

### 4.1 FIC Prior over the State Trajectory

An important alternative to GP-SSM is given by exchanging the vanilla GP prior over $f$ for a sparse counterpart. We do not consider the resulting model to be an approximation to GP-SSM, it is still a GP-SSM, but *with a different prior over functions*. As a result we expect it to sometimes outperform its non-sparse version in the same way as it happens with their regression siblings [16].

Most sparse GP methods can be formulated in terms of a set of so called inducing variables [17]. These variables live in the space of the latent function and have a set $\mathcal{I}$ of corresponding inducing inputs. The assumption is that, conditionally on the inducing variables, the latent function values are mutually independent. Although the inducing variables are marginalized analytically – this is key for the model to remain nonparametric – the inducing inputs have to be chosen in such a way that they, informally speaking, cover the same region of the input space covered by the data. Crucially, in order to achieve computational gains, the number $M$ of inducing variables is selected to be smaller than the original number of data points. In the following, we will use the fully independent conditional (FIC) sparse GP prior as defined in [17] due to its very good empirical performance [16].

As shown in [17], the FIC prior can be obtained by replacing the covariance function $k(\cdot, \cdot)$ by,

$$k^{\text{FIC}}(\mathbf{x}_i, \mathbf{x}_j) = s(\mathbf{x}_i, \mathbf{x}_j) + \delta_{ij}\big(k(\mathbf{x}_i, \mathbf{x}_j) - s(\mathbf{x}_i, \mathbf{x}_j)\big), \tag{7}$$

where $s(\mathbf{x}_i, \mathbf{x}_j) \triangleq k(\mathbf{x}_i, \mathcal{I}) k(\mathcal{I}, \mathcal{I})^{-1} k(\mathcal{I}, \mathbf{x}_j)$, $\delta_{ij}$ is Kronecker's delta and we use the convention whereby when $k$ takes a set as one of its arguments it generates a matrix of covariances. Using the Woodbury matrix identity, we can express the one-step predictive density as in (3), with

$$\boldsymbol{\mu}_t^{\text{FIC}}(\mathbf{x}_{0:t-1}) = \mathbf{m}_{t-1} + \mathbf{K}_{t-1,\mathcal{I}} \mathbf{P} \mathbf{K}_{\mathcal{I},0:t-2} \boldsymbol{\Lambda}_{0:t-2}^{-1} (\mathbf{x}_{1:t-1} - \mathbf{m}_{0:t-2}), \tag{8a}$$

$$\boldsymbol{\Sigma}_t^{\text{FIC}}(\mathbf{x}_{0:t-1}) = \widetilde{\mathbf{K}}_{t-1} - \mathbf{S}_{t-1} + \mathbf{K}_{t-1,\mathcal{I}} \mathbf{P} \mathbf{K}_{\mathcal{I},t-1}, \tag{8b}$$

where $\mathbf{P} \triangleq (\mathbf{K}_{\mathcal{I},\mathcal{I}} + \mathbf{K}_{\mathcal{I},0:t-2} \boldsymbol{\Lambda}_{0:t-2}^{-1} \mathbf{K}_{0:t-2,\mathcal{I}})^{-1}$, $\boldsymbol{\Lambda}_{0:t-2} \triangleq \text{diag}[\widetilde{\mathbf{K}}_{0:t-2} - \mathbf{S}_{0:t-2}]$ and $\mathbf{S}_{\mathcal{A},\mathcal{B}} \triangleq \mathbf{K}_{\mathcal{A},\mathcal{I}} \mathbf{K}_{\mathcal{I},\mathcal{I}}^{-1} \mathbf{K}_{\mathcal{I},\mathcal{B}}$. Despite its apparent cumbersomeness, the computational complexity involved in computing the above mean and covariance is $\mathcal{O}(M^2 t)$, as opposed to $\mathcal{O}(t^3)$ for (3). The same idea can be used to express (4) in a form which allows for efficient computation. Here diag refers to a block diagonalization if $\mathbf{Q}$ is not diagonal.

We do not address the problem of choosing the inducing inputs, but note that one option is to use greedy methods (e.g. [18]). The fast forward selection algorithm is appealing due to its very low computational complexity [18]. Moreover, its potential drawback of interference between hyper-parameter learning and active set selection is not an issue in our case since hyper-parameters will be fixed for a given run of the particle filter.

### 4.2 Implementation Details

As pointed out above, it is crucial to reuse computations across time to attain the $\mathcal{O}(T^3)$ or $\mathcal{O}(M^2 T)$ computational complexity for the vanilla GP and the FIC prior, respectively. We start by discussing the vanilla GP and then briefly comment on the implementation aspects of FIC.

There are two costly operations of the CPF-AS algorithm: *(i)* sampling from the prior (5), requiring the computation of (3b) and (3c) and *(ii)* evaluating the ancestor sampling probabilities according to (6). Both of these operations can be carried out efficiently by keeping track of a Cholesky factorization of the matrix $\widetilde{\mathbf{K}}(\{\mathbf{x}_{0:t-1}^i, \widetilde{\mathbf{x}}_{t:T-1}\}) = \mathbf{L}_t^i \mathbf{L}_t^{i\top}$, for each particle $i = 1, \ldots, N$. Here, $\widetilde{\mathbf{K}}(\{\mathbf{x}_{0:t-1}^i, \widetilde{\mathbf{x}}_{t:T-1}\})$ is a matrix defined analogously to $\widetilde{\mathbf{K}}_{0:T-1}$, but where the covariance function is evaluated for the concatenated state trajectory $\{\mathbf{x}_{0:t-1}^i, \widetilde{\mathbf{x}}_{t:T-1}\}$. From $\mathbf{L}_t^i$, it is possible to identify sub-matrices corresponding to the Cholesky factors for the covariance matrix $\boldsymbol{\Sigma}_t(\mathbf{x}_{0:t-1}^i)$ as well as for the matrices needed to efficiently evaluate the ancestor sampling probabilities (6).

It remains to find an efficient update of the Cholesky factor to obtain $\mathbf{L}_{t+1}^i$. As we move from time $t$ to $t+1$ in the algorithm, $\widetilde{\mathbf{x}}_t$ will be replaced by $\mathbf{x}_t^i$ in the concatenated trajectory. Hence, the matrix $\widetilde{\mathbf{K}}(\{\mathbf{x}_{0:t}^i, \widetilde{\mathbf{x}}_{t+1:T-1}\})$ can be obtained from $\widetilde{\mathbf{K}}(\{\mathbf{x}_{0:t-1}^i, \widetilde{\mathbf{x}}_{t:T-1}\})$ by replacing $n_x$ rows and columns, corresponding to a rank $2n_x$ update. It follows that we can compute $\mathbf{L}_{t+1}^i$ by making $n_x$ successive rank one updates and downdates on $\mathbf{L}_t^i$. In summary, all the operations at a specific time step can be done in $\mathcal{O}(T^2)$ computations, leading to a total computational complexity of $\mathcal{O}(T^3)$.

For the FIC prior, a naive implementation will give rise to $\mathcal{O}(M^2 T^2)$ computational complexity. This can be reduced to $\mathcal{O}(M^2 T)$ by keeping track of a factorization for the matrix $\mathbf{P}$. However, to reach the $\mathcal{O}(M^2 T)$ cost all intermediate operations scaling with $T$ has to be avoided, requiring us to reuse not only the matrix factorizations, but also intermediate matrix-vector multiplications.

## 5 Learning the Dynamics

Algorithm 1 gives us a tool to compute $p(\mathbf{x}_{0:T}, \boldsymbol{\theta} \mid \mathbf{y}_{1:T})$. We now discuss how this can be used to find an explicit model for $f$. The goal of learning the state transition dynamics is equivalent to that of obtaining a predictive distribution over $\mathbf{f}^* = f(\mathbf{x}^*)$, evaluated at an arbitrary test point $\mathbf{x}^*$,

$$p(\mathbf{f}^* \mid \mathbf{x}^*, \mathbf{y}_{1:T}) = \int p(\mathbf{f}^* \mid \mathbf{x}^*, \mathbf{x}_{0:T}, \boldsymbol{\theta}) \, p(\mathbf{x}_{0:T}, \boldsymbol{\theta} \mid \mathbf{y}_{1:T}) \, \mathrm{d}\mathbf{x}_{0:T} \, \mathrm{d}\boldsymbol{\theta}. \tag{9}$$

Using a sample-based approximation of $p(\mathbf{x}_{0:T}, \boldsymbol{\theta} \mid \mathbf{y}_{1:T})$, this integral can be approximated by

$$p(\mathbf{f}^* \mid \mathbf{x}^*, \mathbf{y}_{1:T}) \approx \frac{1}{L} \sum_{\ell=1}^{L} p(\mathbf{f}^* \mid \mathbf{x}^*, \mathbf{x}_{0:T}[\ell], \boldsymbol{\theta}[\ell]) = \frac{1}{L} \sum_{\ell=1}^{L} \mathcal{N}(\mathbf{f}^* \mid \boldsymbol{\mu}^\ell(\mathbf{x}^*), \boldsymbol{\Sigma}^\ell(\mathbf{x}^*)), \tag{10}$$

where $L$ is the number of samples and $\boldsymbol{\mu}^\ell(\mathbf{x}^*)$ and $\boldsymbol{\Sigma}^\ell(\mathbf{x}^*)$ follow the expressions for the predictive distribution in standard GP regression if $\mathbf{x}_{0:T-1}[\ell]$ are treated as inputs, $\mathbf{x}_{1:T}[\ell]$ are treated as outputs and $\mathbf{Q}$ is the likelihood covariance matrix. This mixture of Gaussians is an expressive representation of the predictive density which can, for instance, correctly take into account multimodality arising from ambiguity in the measurements. Although factorized covariance matrices can be pre-computed, the overall computational cost will increase linearly with $L$. The computational cost can be reduced by thinning the Markov chain using e.g. random sub-sampling or kernel herding [19].

In some situations it could be useful to obtain an approximation from the mixture of Gaussians consisting in a single GP representation. This is the case in applications such as control or real time filtering where the cost of evaluating the mixture of Gaussians can be prohibitive. In those cases one could opt for a pragmatic approach and learn the mapping $\mathbf{x}^* \mapsto \mathbf{f}^*$ from a cloud of points $\{\mathbf{x}_{0:T}[\ell], \mathbf{f}_{0:T}[\ell]\}_{\ell=1}^L$ using sparse GP regression. The latent function values $\mathbf{f}_{0:T}[\ell]$ can be easily sampled from the normally distributed $p(\mathbf{f}_{0:T}[\ell] \mid \mathbf{x}_{0:T}[\ell], \boldsymbol{\theta}[\ell])$.

# 6 Experiments

## 6.1 Learning a Nonlinear System Benchmark

Consider a system with dynamics given by $x_{t+1} = ax_t + bx_t/(1 + x_t^2) + cu_t + v_t, v_t \sim \mathcal{N}(0,q)$ and observations given by $y_t = dx_t^2 + e_t, e_t \sim \mathcal{N}(0,r)$, with parameters $(a,b,c,d,q,r) = (0.5, 25, 8, 0.05, 10, 1)$ and a known input $u_t = \cos(1.2(t+1))$. One of the difficulties of this system is that the smoothing density $p(\mathbf{x}_{0:T} \mid \mathbf{y}_{0:T})$ is multimodal since no information about the sign of $x_t$ is available in the observations. The system is simulated for $T = 200$ time steps, using log-normal priors for the hyper-parameters, and the PGAS sampler is then run for 50 iterations using $N = 20$ particles. To illustrate the capability of the GP-SSM to make use of a parametric model as baseline, we use a mean function with the same parametric form as the true system, but parameters $(a,b,c) = (0.3, 7.5, 0)$. This function, denoted *model B*, is manifestly different to the actual state transition (green vs. black surfaces in Figure 2), also demonstrating the flexibility of the GP-SSM.

Figure 2 (left) shows the samples of $\mathbf{x}_{0:T}$ (red). It is apparent that the distribution covers two alternative state trajectories at particular times (e.g. $t = 10$). In fact, it is always the case that this bi-modal distribution covers the two states of opposite signs that could have led to the same observation (cyan). In Figure 2 (right) we plot samples from the smoothing distribution, where each circle corresponds to $(\mathbf{x}_t, \mathbf{u}_t, \mathbb{E}[\mathbf{f}_t])$. Although the parametric model used in the mean function of the GP (green) is clearly not representative of the true dynamics (black), the samples from the smoothing distribution accurately portray the underlying system. The smoothness prior embodied by the GP allows for accurate sampling from the smoothing distribution even when the parametric model of the dynamics fails to capture important features.

To measure the predictive capability of the learned transition dynamics, we generate a new dataset consisting of $10\,000$ time steps and present the RMSE between the predicted value of $f(\mathbf{x}_t, \mathbf{u}_t)$ and the actual one. We compare the results from GP-SSM with the predictions obtained from two parametric models (one with the true model structure and one linear model) and two known models (the ground truth model and model B). We also report results for the sparse GP-SSM using an FIC prior with 40 inducing points. Table 1 summarizes the results, averaged over 10 independent training and test datasets. We also report the RMSE from the joint smoothing sample to the ground truth trajectory.

Table 1: RMSE to ground truth values over 10 independent runs.

| RMSE | prediction of $\mathbf{f}^*\|\mathbf{x}_t^*, \mathbf{u}_t^*, \mathrm{data}$ | smoothing $\mathbf{x}_{0:T}\|\mathrm{data}$ |
|---|---|---|
| Ground truth model (known parameters) | – | $2.7 \pm 0.5$ |
| GP-SSM (proposed, model B mean function) | $1.7 \pm 0.2$ | $3.2 \pm 0.5$ |
| Sparse GP-SSM (proposed, model B mean function) | $1.8 \pm 0.2$ | $2.7 \pm 0.4$ |
| Model B (fixed parameters) | $7.1 \pm 0.0$ | $13.6 \pm 1.1$ |
| Ground truth model, learned parameters | $0.5 \pm 0.2$ | $3.0 \pm 0.4$ |
| Linear model, learned parameters | $5.5 \pm 0.1$ | $6.0 \pm 0.5$ |

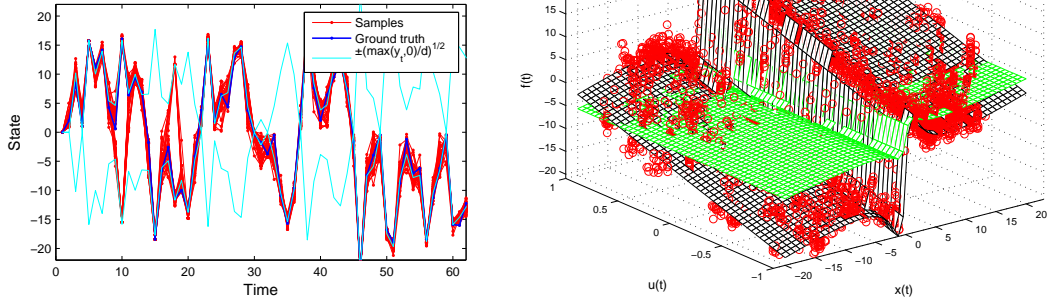

Figure 2: Left: Smoothing distribution. Right: State transition function (black: actual transition function, green: mean function (model B) and red: smoothing samples).

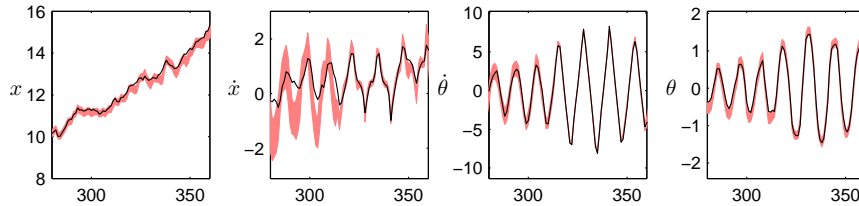

Figure 3: One step ahead predictive distribution for each of the states of the cart and pole system. Black: ground truth. Colored band: one standard deviation from the mixture of Gaussians predictive.

## 6.2 Learning a Cart and Pole System

We apply our approach to learn a model of a cart and pole system used in reinforcement learning. The system consists of a cart, with a free-spinning pendulum, rolling on a horizontal track. An external force is applied to the cart. The system's dynamics can be described by four states and a set of nonlinear ordinary differential equations [20]. We learn a GP-SSM based on 100 observations of the state corrupted with Gaussian noise. Although the training set only explores a small region of the 4-dimensional state space, we can learn a model of the dynamics which can produce one step ahead predictions such the ones in Figure 3. We obtain a predictive distribution in the form of a mixture of Gaussians from which we display the first and second moments. Crucially, the learned model reports different amounts of uncertainty in different regions of the state-space. For instance, note the narrower error-bars on some states between $t = 320$ and $t = 350$. This is due to the model being more confident in its predictions in areas that are closer to the training data.

## 7 Conclusions

We have shown an efficient way to perform fully Bayesian inference and learning in the GP-SSM. A key contribution is that our approach retains the full nonparametric expressivity of the model. This is made possible by marginalizing out the state transition function, which results in a non-trivial inference problem that we solve using a tailored PGAS sampler.

A particular characteristic of our approach is that the latent states can be sampled from the smoothing distribution even when the state transition function is unknown. Assumptions about smoothness and parsimony of this function embodied by the GP prior suffice to obtain high-quality smoothing distributions. Once samples from the smoothing distribution are available, they can be used to describe a posterior over the state transition function. This contrasts with the conventional approach to inference in dynamical systems where smoothing is performed conditioned on a model of the state transition dynamics.

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
