[Reviews · NeurIPS 2013]

Submitted by Assigned_Reviewer_6

This paper presents a Bayesian approach to state and parameter estimation in nonlinear state-space models, while also learning the transition dynamics through the use of a Gaussian process (GP) prior. The inference mechanism is based on particle Markov chain Monte Carlo (PMCMC) with the recently-introduced idea of ancestor sampling. The paper also discusses computational efficiencies to be had with respect to sparsity and low-rank Cholesky updates.

This is a technically sound and strong paper with clear and accessible presentation. The online marginalisation of the transition dynamics and the use of ancestor sampling to achieve this is novel. The consideration of computational issues such as sparsity and low-rank updates/downdates to the Cholesky factors of covariance matrices strengthens the paper further. The empirical results, while brief, are sufficient (further suggestions below).

In addition to its stated aims, the paper will likely stimulate discussion around inference methods for non-Markovian state-space models and the potential advantages/disadvantages of learning the transition dynamics in this way rather than specifying a parametric model a priori.

While space is slight, the authors may like to consider some further discussion around the differences between using a parametric transition model given a priori against the use of a similar model as the mean function of the GP. For example in out of sample prediction (e.g. forecasting).

The results in Table 1 and the description in the preceding paragraph are slightly unclear to me. I am unsure as to whether the RMSE is against a withheld set of data points or the same set of data points that is conditioned upon (the *|data in the column headings). My main interest would be an RMSE against an out-of-sample prediction, especially a forecast forward in time against withheld data. It is in this scenario that I would expect to see the largest differences between the learnt dynamics and the ground truth model. If Table 1 is not already showing this, an extra column that does so would be a great addition.

One minor point: the abbreviation CPF-AS is used in Algorithm 1 before being defined in the first paragraph of Section 3.3.1 below.
Summary: A strong and novel paper that should stimulate some interesting discussion.

Submitted by Assigned_Reviewer_9

The authors propose to apply particle MCMC to perform inference in Gaussian process state-space models. In particular, they focus on the recent ancestral sampling particle Gibbs algorithm of Lindsten et al. The paper is clear and it is an interesting and original application of particle MCMC. There are also some useful model-specific methodology developed in the paper, namely sparse GP-SSM.

One thing I find truly regrettable is the lack of comparisons to other particle MCMC schemes, in particular the particle marginal MH (PMMH) scheme and the particle Gibbs with backward sampling (as in Whiteley et al.). They could have been straightforwardly implemented and it would be of interest to know how those variants compared to the proposed scheme (and it would not be much work for the authors either).

Additionally I would like to see graphs displaying the performance of the algorithms (e.g. in terms of ACF or ESS) as a function of N and T. As they stand the results are not very informative. Do I need to scale N linearly with T, sublinearly?
I believe that for such models the PMMH would require a number of particles increasing quadratically with T as observed in Whiteley et al. whereas both particle Gibbs require a number of particle growing sublinearly with T.



Summary: A well-written application of particle MCMC to GP state-space models. The paper could be significantly improved if the proposed algorithm was compared to the PMMH and the particle Gibbs with backward sampling.

Submitted by Assigned_Reviewer_10

PMCMC sampling is exploited in an ssm with GP process prior to extend to actual parameters rather than just the usual filtering and smoothing. Nice straightforward application of PMCMC methodology. Pity that a proper evaluation of the challenge tp get the PMCMC scheme to work is not described and evaluated in more detail as this would really have been the important contribution. A more detailed and critical evaluation of the strengths and weaknesses of the approach would have made the paper of value given it is an application of PMCMC methodology. It is probably too much to ask that the experimental section be revised to provide more evaluation than demonstration.
Summary: Application of PMCMC methodology to ssm

would have been more useful assessing the practicial difficulties in getting such a scheme to work - and how well it actually works
Author Feedback

Author rebuttal: We thank the reviewers for their positive comments.

We would like to emphasize that PMCMC has allowed us to learn Bayesian GP state-space models while keeping alive the whole nonparametric richness of this kind of model. We believe that it is the first time that this has been achieved irrespective of the inference method used. In our opinion, this is a valuable result in Bayesian Nonparametrics in its own right besides being a demonstration of the power of PMCMC.

The practical difficulties in getting PMCMC to work efficiently were solved through the marginalization of the latent function f(x), the use of a sparse covariance function (FIC) and the careful sequential construction of Cholesky factorizations of the covariance matrix. Those (non trivial to us) contributions allowed Particle Gibbs with Ancestor sampling (PGAS) to perform very well "out of the box". In our opinion this was possible thanks to: 1) all the work done in adapting the model for efficient sampling and 2) the inherent power of PGAS to sample from non-Markovian models such as the one induced by the marginalization of the GP.

We agree that providing more experimental evaluation would improve the paper. However, severe space limitations did not allow us to present in much detail very important computational aspects of our method such as the sparse GP-SSM or the sequential updating of factorizations of covariance matrices. As a consequence, we felt that the small amount of space left would be better used in providing an illustrative demonstration of the capabilities of our approach to Bayesian inference in GP-SSMs. In particular, the figures emphasized the particles from the smoothing distribution since an unconventional property of this state-space model is that any prediction made by the model uses the particles of the smoothing distribution. This is in contrast with parametric models where the learned parameters contain all that is needed to make predictions.

Although comparison with other PMCMC methods would undoubtedly make the paper stronger, our choice of Particle Gibbs with Ancestor sampling (PGAS) was motivated by its particularly good performance for non-Markovian models such as the one obtained when marginalizing the Gaussian process latent function. In the original PGAS paper (Lindsten, Jordan and Schön, 2012), the authors showed how PGAS consistently performed better than Particle Gibbs with Backward Simulation when applied to non-Markovian models.

The RMSEs reported in Table 1 are indeed out-of-sample predictions on long data records. We will make sure to update the text to try to remove any ambiguity regarding these out-of-sample predictions.